# Cementless Metal-Free Ceramic-Coated Shoulder Resurfacing

**DOI:** 10.3390/jpm13050825

**Published:** 2023-05-13

**Authors:** James W. Pritchett

**Affiliations:** Swedish Medical Center, 901 Boren Ave., Suite 711, Seattle, WA 90104, USA; bonerecon@aol.com; Tel.: +206-779-2590

**Keywords:** shoulder resurfacing, arthroplasty, cementless resurfacing, ceramic-coated implant

## Abstract

Shoulder resurfacing is a versatile, bone-conserving procedure to treat arthritis, avascular necrosis, and rotator cuff arthropathy. Shoulder resurfacing is of interest to young patients who are concerned about implant survivorship and those in need of a high level of physical activity. Using a ceramic surface reduces wear and metal sensitivity to clinically unimportant levels. Between 1989 and 2018, 586 patients received cementless, ceramic-coated shoulder resurfacing implants for arthritis, avascular necrosis, or rotator cuff arthropathy. They were followed for a mean of 11 years and were assessed using the Simple Shoulder Test (SST) and Patient Acceptable Symptom State (PASS). CT scans were used in 51 hemiarthroplasty patients to assess the glenoid cartilage wear. Seventy-five patients had a stemmed or stemless implant in the contralateral extremity. A total of 94% of patients had excellent or good clinical results and 92% achieved PASS. 6% of patients required a revision. A total of 86% of patients preferred their shoulder resurfacing prosthesis over a stemmed or stemless shoulder replacement. The glenoid cartilage wear at a mean of 10 years was 0.6 mm by a CT scan. There were no instances of implant sensitivity. Only one implant was removed due to a deep infection. Shoulder resurfacing is an exacting procedure. It is clinically successful, with excellent long-term survivorship in young and active patients. The ceramic surface has no metal sensitivity, very low wear, and, therefore, it is successful as a hemiarthroplasty.

## 1. Introduction

The humeral head has been considered a surplus skeletal part and is removed routinely by most surgeons when performing a shoulder implant procedure. Replacement with a metal head attached to a stem placed into the medullary canal of the humerus has been the chosen reconstructive treatment. This may be an unnecessary concession to convention and surgeon’s convenience [1,2,3,4]. It may not be the best option for every patient. During the initial development of implant procedures, surgeons were concerned about the intrusive nature of the procedures and the amount of metal involved. The biocompatibility of the implant itself was an additional concern. Any implanted material must be durable, capable of excellent functional performance, and biocompatible.

Cup arthroplasty was suggested initially for both the hip and shoulder [5,6,7]. Resurfacing arthroplasty, or surface replacement, is an evolved technique from cup arthroplasty. In this technique, only the degraded surface of the humeral head is replaced and, when necessary, the glenoid is resurfaced. Although it is easier and less demanding to excise the humeral head and perform a total joint replacement, a resurfacing procedure is more conservative and can be a better option for the patient [8,9,10,11,12,13,14,15,16,17].

Humeral head resurfacing is best suited for a young patient with arthritic involvement predominately of the humeral head. It can also be used to treat avascular necrosis and the occasional head-splitting fracture. There are cases in which resurfacing is the best option because there is a deformity or prior surgery with existing hardware in the humerus blocking the placement of a stemmed prosthesis. It can be used in teenagers or in cases with an elevated fear of infection. Contraindications to resurfacing include poor bone quality or unstable soft tissues. The under surface of the implant is coated with a porous surface for ingrowth/ongrowth [8,9,12,13,14,15].

In the majority of cases, the humeral resurfacing prosthesis will articulate directly with the preserved glenoid cartilage [8,9,10,11,12,13,14,15]. Therefore, the smoothest possible surface is necessary. Ceramic coating the implant is the best method to achieve this goal (Figure 1). 

Titanium nitride (TiN) has high hardness and a low friction coefficient. It has been shown to be efficient in reducing the wear of cutting tools. It has been cleared by the Food and Drug Administration as an implant bearing surface and coating (510K 93122). It has much less wear compared to the cobalt chromium implants that are typically used for shoulder replacement and other resurfacing implants. In addition to reduced wear, there are no metal ions released into the tissues and, therefore, no chance of metal sensitivity or an allergic reaction [18,19,20]. The glenoid component, when needed, is either cemented or porous-backed polyethylene [10,15] (Figure 2).

This study answered five questions:Does cementless, ceramic-coated shoulder resurfacing produce satisfactory function and survivorship?What is the wear of ceramic resurfacing prostheses?Can a ceramic humeral resurfacing prosthesis produce acceptable outcomes without glenoid resurfacing?Do patients prefer shoulder resurfacing to other implant procedures?Can a ceramic humeral shoulder resurfacing prosthesis provide acceptable outcomes for rotator cuff arthropathy patients?

## 2. Materials and Methods

The Institutional Review Board approved this retrospective study. The patients reported in this study underwent their shoulder procedures between 1989 and 2018. The patients’ function and comfort were assessed using the Simple Shoulder Test (SST) before surgery and at follow-up every other year [4,21]. Patients were followed in person, with video visits, and electronically. We used the SST and the Patient Acceptable Symptom State (PASS) because they allow patients to assess their own function and comfort. The most recent SST score was compared with the preoperative score to assess the outcome. In addition, patients who had a stemmed or stemless prosthesis in their other shoulder were asked which shoulder they preferred. They were also asked the reasons for the preference. Most of our patients were from out of town with an average distance from the clinic of 1160 miles. There were 3 groups of patients:Humeral hemiarthroplasty.Total resurfacing arthroplasty with a ceramic humeral resurfacing component articulating with a polyethylene glenoid.Hemiarthroplasty for rotator cuff arthropathy.

Shared decision-making was used to determine the procedure that a patient preferred. A stem-supported anatomic or reverse total shoulder replacement was performed according to the patient’s choice. Additionally, a glenoid component was placed at the time of humeral head resurfacing if there was significant glenoid wear or if this was the patient’s choice.

All patients received radiographs. The radiographs were examined to determine component fixation, component position, bone loss, and, if evident, glenoid cartilage wear. A total of 51 hemiarthroplasty patients received CT scans 9–11 years after their surgery to accurately measure the glenoid cartilage wear [22].

Retrieval analysis was performed on implants that were obtained either postmortem or if there was a revision.

### 2.1. The Humeral Head Implant

The titanium alloy Ti6Al_4_V humeral implant was straight-stemmed and proportional from 44–56 mm in diameter. Its undersurface was porous-coated with commercially pure titanium plasma spray/beads to give an average pore size of 350 µm and a volume porosity of 30%. The articulating surface was made of titanium alloy coated with a polished 10 µm-thick layer of titanium nitride ceramic (TiN). This polishing was to 0.03 µ before and to 0.04 µ after the TiN coating. The TiN coating was applied using the Physical Vapor Deposition process [18]. When a glenoid component was used, it was Ethylene Oxide sterilized GUR 1020 polyethylene.

### 2.2. Surgical Technique

The resurfacing arthroplasty technique was designed to restore anatomy to the articular surface of the humerus by applying a “cap” over the reamed surface of the humeral head. An anatomic position restored the normal degree of retroversion (average 30°) and valgus (average 140°). This allowed the tuberosities to be maintained with the rotator cuff attachments and a preserved force couple.

When there is rotator cuff tear arthropathy with superior migration of the humeral head but with a still captured humeral head within the coracoacromial arch, resurfacing is possible. An “acetabularized” articulation with the humeral head using both the glenoid and acromion is created. A now smooth surface can allow this articulation with altered mechanics to provide reasonable function while reducing pain. In this situation, the direction of the humeral head reaming may be as high as 170° valgus (“hyper valgus”) and a femoral rather than humeral resurfacing device. These are limited goal cases from a functional standpoint. The arm can be raised to eye level.

For both anatomic and rotator cuff arthropathy resurfacing, the operative technique is similar: a deltopectoral approach is used. A limited number of pectoralis fibers can be released to visualize the subscapularis. In cases of cuff tear arthropathy, the upper subscapularis may have eroded; the supraspinatus and infraspinatus are usually retracted medial to the humeral head. The humeral head rides high in the subacromial space but is contained within the subacromial arch. In the more typical cases with an intact or nearly intact rotator cuff, the subscapularis is incised with a stump for later repair. The dissection stops at the rotator cuff interval, avoiding injury to the insertion of the supraspinatus and then the head is dislocated anteriorly. In all cases, great care is taken to preserve the coracoacromial ligament.

Once the humeral head has been delivered into the wound, the well-visualized peripheral osteophytes are removed. A center guide wire is placed at the normal inclination of the humeral head, perpendicular to the apex of the natural articular surface. In cases of cuff tear arthropathy, the guide wire is placed in hyper valgus, which will allow seating over the entire superior humeral surface including the tuberosities. If the biceps are healthy, it is left intact; if not, a tenodesis is performed as part of the closure. The head sizer is placed over the guide wire. Caution should be taken not to start the reamer before full application to the bone to avoid grabbing and fracturing the humeral head or neck. Only the articular surface is reamed down to bleeding bone. 

### 2.3. Statistical Analysis

The clinical assessments were based on written responses to the SST [4,21]. These responses were collected through January 2018, preoperatively and postoperatively. The percentage of maximum possible improvements was calculated for the SST score with the following formula: SST total score at the time of follow-up—SST total score preoperatively × 100%/12 points—SST total score preoperatively.

Patients were considered to have achieved a meaningful improvement if the SST increased by at least 30% of the maximum possible improvement. This method avoids the ceiling effect that results from defining minimum clinically important differences. Survivorship was defined as the absence of revision procedures and calculated using a Kaplan–Meier estimator.

The PASS test was used as the more sensitive measure to assess the outcome in this unique and demanding population. The PASS question used was as follows: “Taking in account your shoulder pain and function and how it affects your daily life including your ability to participate in sport and social activities, do you consider your current state acceptable?” PASS determines if a patient improves to the point of getting well.

## 3. Results

The clinical outcomes were reported for three groups of patients:Resurfacing hemiarthroplasty.Total shoulder resurfacing with a polyethylene glenoid.Resurfacing arthroplasty for rotator cuff arthropathy.

### 3.1. Shoulder Resurfacing Hemiarthroplasty

There were 428 patients: 286 (67%) were men, 137 (32%) were women, and 4 (1%) were non-binary. The mean patient age was 52 (standard deviation SD 10.2 years; range, 15–73 years). A total of 158 (37%) had prior surgery on their shoulder. The mean preoperative SST score was 4 (SD 2.5) out of a possible 12 shoulder functions. For the unrevised shoulder followed for a minimum of 5 years (mean 11 years; range 5–30 years), the mean SST score was 10.5 (SD 2.9) of 12 possible positive responses. The median SST was 11 points (interquartile range, 9–12 points). A total of 402 (94%) hemiarthroplasty patients obtained ≥ 30% of the maximum possible improvement in the SST score between the preoperative and peak evaluation. A total of 92% of patients achieved PASS (Table 1). A total of 26 (6%) hemiarthroplasty patients had subsequent procedures: 9 (2%) hemiarthroplasty patients had a revision to add a glenoid component, 5 had revisions to total shoulder replacement, 5 had revisions to a reverse total shoulder replacement, 3 had subscapularis or rotator cuff repairs, 2 had fracture repairs, and 2 had arthroscopic release procedures. There were twenty-three incision infections with three deep infections. There was one dislocation and two fractures. There were five brachial plexopathies and one had a permanent partial median nerve deficit. There was one loose humeral resurfacing implant.

### 3.2. Total Shoulder Resurfacing

There were 91 patients treated with a total resurfacing shoulder: 51 (56%) were men and 40 (44%) were women. The mean patient age was 64 (SD 10.3 years; range 39–77). A total of 35 (39%) had prior surgery. The mean preoperative SST score was 4 (SD 2.5). For the total shoulder arthroplasty, the mean postoperative SST score was 9.9 (SD 3.1) of 12 possible responses. The median SST was 11 points (interquartile range; 9–12 points). A total of 74 patients (90%) obtained ≥ 30% of the maximum possible improvement in the SST score and 90% achieved PASS (Table 1). A total of nine (12%) total shoulder resurfacing patients had revision procedures: three were to a reverse total shoulder arthroplasty, three were glenoid revisions for loosening (in one instance, the glenoid prosthesis was removed and not replaced), two were rotator cuff repairs, and one was an arthroscopic lysis of adhesions. There were two incision infections and no deep infections. There was one brachial plexopathy that resolved.

### 3.3. Rotator Cuff Resurfacing Arthropathy

For the 67 rotator cuff arthropathy patients, the mean patient age was 67 (SD 10.6 years, range; 51–83 years): 46 (68%) were men, 21 (32%) were women, and 44 (67%) had prior surgery (Figure 3). 

The mean preoperative SST score for rotator cuff arthropathy was 2.5 (SD 2.5) out of a possible 12. For the rotator cuff arthropathy patients, the mean SST score was 7.7 (SD 4) of 12 possible responses. The median SST was 8 (interquartile range; 5–11 points). A total of 43 (64 %) obtained ≥ 30% of the maximum possible improvement in the SST score. A total of 64% achieved PASS. There were three incision infections and one deep infection in a multiply operated shoulder. There were three brachial plexopathies and one had a permanent medial nerve deficit. Of the combined group, 31 patients were lost to follow-up and the outcomes at a minimum of 5 years were reported. Nineteen patients died of causes unrelated to shoulder resurfacing. The number of deaths was less than the actuarial predictions for the general population and were not shared by total shoulder replacement. This was also reported for hip resurfacing [23].

### 3.4. Patient Preference

A total of 55 resurfacing hemiarthroplasty patients had a different prosthesis on the other side: a stemmed total shoulder replacement (19), stemmed hemiarthroplasty (18), total shoulder resurfacing (9), reverse shoulder arthroplasty (7), and resurfacing hemiarthroplasty for rotator cuff arthropathy (2). A total of 11 shoulder arthroplasty patients had a different prosthesis on the other side: stemmed total shoulder replacement (5), stemmed hemiarthroplasty (3), resurfacing hemiarthroplasty (2), and reverse total shoulder arthroplasty (1). A total of 9 rotator cuff arthropathy patients had a different prosthesis on the other side: stemmed total shoulder arthroplasty (3), reverse total shoulder arthroplasty (2), stemmed hemiarthroplasty (2), and resurfacing hemiarthroplasty (2).

A total of 50 out of 55 (90%) of hemiarthroplasty patients preferred their hemiarthroplasty compared to their other shoulder. Three preferred their total resurfacing shoulder or stemmed total shoulder and two had no preference. Eight of the eleven total shoulder resurfacing patients preferred their total shoulder resurfacing to their other shoulder. Two preferred their resurfacing hemiarthroplasty and one had no preference. Six out of nine rotator cuff arthropathy patients preferred their hemiarthroplasty to their other prosthesis, and three preferred their reverse total shoulder arthroplasty (Table 2). The reasons provided were consistent: it feels more natural, it feels more stable, and I can do more with it [24,25]. The operative time for resurfacing hemiarthroplasty was a mean of 24 min longer compared to stemmed hemiarthroplasty. No patient had signs or symptoms of metal sensitivity. Titanium ion concentrations were measured in 39 patients and they were all unmeasurable, indicating low wear and bonding of the ceramic coating.

### 3.5. Radiographic Examination

Radiographs showed the humeral resurfacing implant was placed correctly in all cases. There were no implants with loosening or radiolucent lines. There were no areas of osteolysis. The glenohumeral joint space, as visualized on plain films, was maintained. The glenohumeral joint space was measured by a CT scan in 51 hemiarthroplasty patients aged 9–11 years (SD 3 years, range 5–15 years) following surgery. There was a mean decrease of 0.6 mm (range, 0–1.6). There were 12 cases where patients started with < 2 mm of joint space.

### 3.6. Retrieval Studies

A postmortem or revision retrieval analysis was available for nine TiN-coated humeral implants obtained 10 or more years following implantation. The average TiN wear at the head pole of the spherical surface was 1.5 µm, with the remaining 8 µm intact. There were four retrievals of polyethylene glenoid components articulating with TiN humeral components. The maximum polyethylene volumetric wear was 19 mm^3^/year. There were no areas of polyethylene wear through. There were no instances of reactive synovitis or osteolysis.

## 4. Discussion

Shoulder resurfacing is an effective procedure for increasing function and reducing pain. It has a higher PASS score compared to stemmed supported implants. It is effective both for high-demand patients with an intact rotator cuff as well as for limited-goal patients with rotator cuff arthropathy. Complications from shoulder resurfacing are less frequent and less serious compared to stemmed hemiarthroplasty and stemmed or stemless total shoulder replacement. Only one shoulder resurfacing implant was explanted due to an infection. The other infections were resolved or suppressed with antibiotics. Resurfacing implant survivorship is also better than for stemmed total shoulder replacement. Shoulder resurfacing “burns no bridges”. Shoulder resurfacing is a very valuable procedure when the humeral canal is blocked, or when there is an elevated concern for infection.

Revision to stemmed total shoulder replacement or reverse total shoulder replacement was uncommon but uncomplicated and successful when necessary [5,6,8,9,10,12,13,14,15,16,17,24]. Shoulder resurfacing is as successful as hemiarthroplasty because glenoid wear is low.

Shoulder resurfacing is a safer and less intrusive procedure compared to total shoulder replacement or hemiarthroplasty with a stemmed or stemless implant. Humeral head resurfacing is also more effective and protective than the “ream and run” procedure. In the “ream and run” procedure, the glenoid is made smooth by reaming but no glenoid implant is placed [4,21,26,27,28]. A stemmed humeral implant with a prosthetic humeral head articulates with the prepared glenoid. There are three reasons why cementless ceramic shoulder resurfacing is a better solution than “ream and run”:All the bone is preserved, and the anatomy is restored more precisely. The exact dimension of the humeral head can be recovered, and the retroversion of the shoulder is restored to the anatomic position [29].A humeral stem is avoided. Stems create an abnormal load transfer across the shoulder joint with stress shielding of the humerus. Additionally, implantation of a stem increases the difficulties involved if a prosthetic infection occurs. Stemless implants are similar to stemmed implants in that they also produce abnormal forces on the proximal humerus [30,31]. While convenient, stemless implants are not equivalent to head-conserving resurfacing implants.A ceramic surface is biocompatible and produces less tissue reaction and wear compared to cobalt chromium. [32]

For patients with rotator cuff arthropathy, shoulder resurfacing offers a less intrusive, lower risk approach than reverse total shoulder replacement. In the event of a reverse total shoulder replacement failing, reconstruction can be very difficult. The results from reverse total shoulder replacement are quite satisfactory and often better than the limited goals of shoulder resurfacing, but the risks are much greater than other shoulder implant procedures [5,27,33].

Patients prefer cementless, ceramic-coated shoulder hemiarthroplasty over total shoulder replacement, reverse total shoulder replacement, and stemmed shoulder hemiarthroplasty. The reasons are a more natural feel and better function. Preference studies are the best way to make comparisons, as all the other variables are controlled [25]. Preference studies have been useful for controlling bias and they have become an accepted method of making comparisons.

It is important to use PASS in assessing the outcome from surgery, as other Patient-Reported Outcome Measures (PROMs) provide some useful information but have their limitations, particularly in highly active individuals. Additionally, survivorship is not the same as a good outcome, and getting better is not equivalent to feeling well. PASS is a more useful method for determining the benefit of a procedure. Achieving PASS may still not reflect a normal state of function. It remains important for both patients and surgeons to recognize that there are limits in our ability to restore shoulder function and measure outcomes. PASS is achieved more often with resurfacing compared to stemmed procedures.

It is very important to use a metal-free implant. A wear simulation test was conducted using 12 specimens to 10 million cycles. The maximum wear patch in the TiN was 1.5 µm compared to the pretest thickness of the coating of 10 µm. The wear penetration, changes to surface roughness, and wear particle analysis showed more than 30 times less wear with ceramic-coated titanium compared to cobalt chromium [34].

Both the retrieval and wear simulator analyses show that a lifetime of use is expected with a ceramic surface. Ceramic wear is much less compared to cobalt chromium. Even without functional or radiographic failure, cobalt chromium wear debris produces synovitis, which can be painful. TiN particles are biocompatible and result in an increase in the proliferation of cells and the affinity of bone to the implant. Cobalt-chromium particles cause osteolysis and chondrolysis. There is lower adhesion of bacteria to TiN surfaces compared to coated or uncoated titanium, cobalt chromium, and polyethylene surfaces and, therefore, a lower infection rate [20,32,33,35,36].

Infections are an important patient concern following any implant arthroplasty procedure. A total of 14.3% of patients raised concerns about incisional infections. Among the patients who were not concerned, the deep prosthetic infection rate was 0.7%. Among patients who had some concerns but no definite superficial infection, the deep infection rate was 3%. For patients with a definite superficial infection, 30% developed a deep prosthetic infection [37]. Infection with resurfacing is less common and much easier to deal with as the medullary space has not been entered. Implant retention is more likely with resurfacing compared to stemmed implant procedures. Only one resurfacing implant was removed for infection, but more than half of the infected stemmed shoulder prostheses were removed.

The results of this study should be reviewed in light of certain limitations. First, all procedures were performed by a surgeon experienced in the technique. Shoulder resurfacing is a demanding technique requiring a close match of the implant to the reshaped native bone. Second, only a single cementless, ceramic-coated implant was used. This implant meets demanding polishing and coating specifications; other implants might not perform as well. Third, most of the patients lived at a distance from the clinic, so we relied on the patients’ assessments of their own shoulder functions and comfort using the SST and PASS. Fourth, this procedure was offered to highly motivated patients who understood that their rehabilitation might be long and challenging. Part of the motivation of the patients was the potential to return to a higher level of physical activity. Fifth, there may be patient and surgeon bias in favor of resurfacing due to its bone-conserving nature. Patients are appreciative of the aesthetics of the implant. For the surgeon, however, resurfacing procedures are demanding and take additional time to perform, but since they are hemiarthroplasty procedures, they are compensated at a lower rate compared to total shoulder replacement. For the patient, the expectations are higher, so the outcomes temper the positive bias.

## 5. Conclusions

In conclusion, cementless, ceramic-coated shoulder resurfacing is a valuable procedure. It has conceptual, procedural, functional, wear, and preference advantages over conventional shoulder replacement options in treating advanced articular cartilage damage in the shoulder.

## Figures and Tables

**Figure 1 jpm-13-00825-f001:**
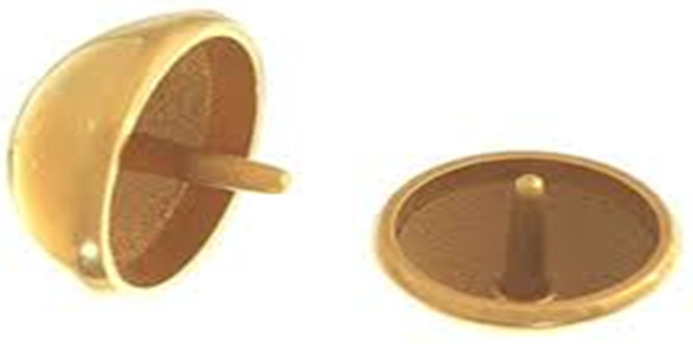
Cementless, ceramic-coated shoulder resurfacing implant.

**Figure 2 jpm-13-00825-f002:**
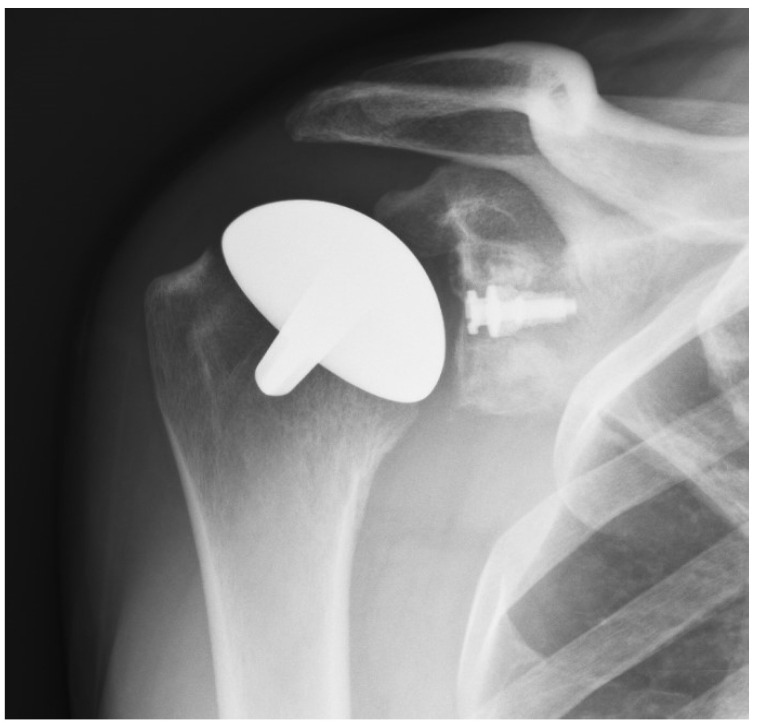
AP radiograph of a shoulder resurfacing with a polyethylene glenoid resurfacing component.

**Figure 3 jpm-13-00825-f003:**
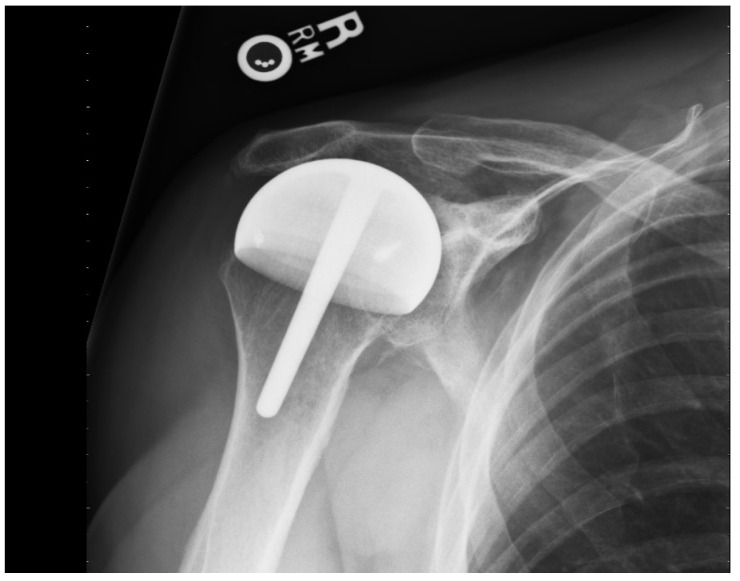
This AP shoulder radiograph shows a shoulder resurfacing performed with a full coverage component covering the tuberosities and articulating with the acromion and glenoid.

**Table 1 jpm-13-00825-t001:** Results by procedure type.

Study Procedure	SST (% Good or Excellent Results)	PASS (% Achieved)	Prosthesis Revision (%)	Complications (*n*)
Resurfacing Hemiarthroplasty	94	92	6	7
Resurfacing Total Arthroplasty	90	90	12	3
Stemmed Hemiarthroplasty	80	81	15	12
Stemmed Total Arthroplasty	84	86	12	11
Resurfacing Cuff Arthropathy Arthroplasty	66	76	6	7
Stemmed Reverse Total Shoulder Arthroplasty	69	77	11	13

**Table 2 jpm-13-00825-t002:** Patient preferences comparison.

Study Procedure	Study Procedure vs. Prosthesis in Contralateral Shoulder		
	Prefer Resurfacing	Prefer Stemmed	No Preference
Resurfacing Arthroplasty vs. Stemmed Total Arthroplasty Resurfacing Hemiarthroplasty vs. Stemmed Hemiarthroplasty Resurfacing Arthroplasty for Arthropathy vs. Stemmed Total Arthroplasty	90% 70% 80%	5% 30% 20%	5% 0% 10%

## Data Availability

The data generated and analyzed in this study are included in this published article.

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
