# Peer review of "Cementless Metal-Free Ceramic-Coated Shoulder Resurfacing"

_jpm, 2023, doi:10.3390/jpm13050825_

Round 1
Reviewer 1 Report
Thank you for giving me the opportunity to read your paper dealing with ceramic coated humeral head resurfacing. Overall, you present a heterogenous population (HA, TSA, CTA). The majority of the patients were treated with resurfacing HA. The results indicate high patient satisfaction and low complication rates after 11 years. Furthermore, cartilage wear seems low in CT scans.
I am afraid that the paper seems to be overloaded by heterogenous indications and surgical procedures. I can understand that you want to present all results, clinical and biomechanical. However, I would recommend shortening the paper. Cyclic loading and wear could be presented in an extra paper and mentioned in the discussion. I would furthermore exclude CTA. This could be a way to create a shorter paper.
Abstract:
The Abstract is a bit confusing… 586 patients, 51 patients, 75 patients… CT scans only in HA patients? When was implant testing carried out?
Introduction:
Overall, well written. I recommend defining your scientific goals of your study (hypothesis).
Line 65: Cemented PE? Hybrid PE?
Line 71-79: Did your study answer these questions, or were they rather a hypothesis?
Material and methods:
Precise and comprehensive.
Results:
Results displayed neatly. There is a broad number of results dealing with HA, TSA, and HA for CTA.
Additionally cyclic wear simulation and retrieval studies are presented.
I have no objections concerning the results, however this part seems almost overloaded.
It is quite amazing that no radiographic findings like stress-shielding, radiolucency or signs of radiologic loosening were observed after 11 years. Did you observe radiolucency at the glenoid in cases of TSA?
Discussion:
Overall, you had a limited number of TSA in comparison to resurfacing? Is this a routine in your clinic? Is the surgical procedure of resurfacing associated to lower rates of glenoid replacement due to a more demanding procedure?
Author Response
JPM 2366468
Response to Reviewer #1
Dear Editors:
Thank you for the comments and suggestions. They have been included in the rewritten and revised manuscript. Every question has been addressed. The critical review and addressing each concern have strengthened the message of the paper.
- The paper seems overloaded. Cyclic loading and wear could be addressed in an extra paper and mentioned in the discussion.
Answer. The cyclic loading and wear information has been removed from the Abstract, Methods, and Results. Just 2 lines of the this are mentioned in the Discussion, line 332.
- The cartilage wear seems low and the majority of patients were hemiresurfacing.
Answer. The wear is low. This is one of the drivers of reporting this information in a paper. The
ceramic coating is the explanation, as the wear with ceramic is 30 times lower compared to
cobalt chromium. Also, resurfacing implants can match normal anatomy more closely
compared to stem implants.
- The reviewer asks about excluding the rotator cuff arthropathy results.
Answer. Reverse Total Shoulder (RSA) is now the most common form of shoulder implant arthroplasty in our area. RSA is growing rapidly and patients ask about this commonly
now. It is important to make information about alternatives to RSA available and, therefore,
This shoulder procedure should still be included.
- The Abstract is confusing. CT scans only in HA patients. When was implant testing carried out.
Answer. The abstract has been rewritten and shortened. CT scans were taken at a mean of 10 years after implantation. CT was used to assess both cartilage and polyethylene wear. Also, implant loosening and bone loss can be identified by CT.
- Recommend defining the scientific goals. Did the study answer the questions posed or are these hypotheses? Were glenoid components when used cemented or hybrid.
Answer. The goals of the study were the 5 questions asked. All 5 were answered. The hypotheses of the study were the questions. The glenoid components were primarily cementless with a central cementless post or a porous backing. A small amount of cement was used in the peripheral holes of some cases and as shown in the Figure. All humeral components were cementless.
Answer. This section is rewritten and the information describing the wear and simulator testing
Is removed.
- This section seems overloaded.
Answer. The result section is rewritten and shortened. The wear and simulation information is
Removed.
- Did you observe stress shielding or sign of radiolucency? Did you see signs of glenoid radiolucency?
Answer. We did not seen signs of stress shielding or radiolucency around the cementless ceramic coated humeral resurfacing. There was stress shielding and radiolucency seen around
the comparison total shoulder replacements including the glenoid components but this was
not a focus of the paper. The glenoid components when used with resurfacing humeral components did not show signs of radiolucency.
- The reviewer notes that a limited number of TSA procedures are reported in comparison to resurfacing. The reviewer also asks is this routine in our clinic? The reviewer also asks if the demanding procedure of placing a glenoid component during a shoulder resurfacing is associated with the lower rate of glenoid replacement?
Answer. This manuscript was focused on resurfacing. Stem supported total shoulder replacement cases were not included except for patients who had a resurfacing on the contralateral shoulder. Our clinic performed stem supported total shoulder replacement both using anatomic and reverse configurations. We perform stem support procedures if the humeral head bone is poor in older patients. We also performed stem supported arthroplasty if this is the patient’s choice after shared decision making.
We place a glenoid component at the time of the humeral resurfacing is there is significant or eccentric wear of the native glenoid. We also place a glenoid component if this is the patient’s choice. It is difficult to place a glenoid component during shoulder resurfacing but this was not a barrier or limitation to doing so when indicated.
- The editors asked for the references to be updated to included papers from the last 8 years.
Answer. The references have been updated with newer references when possible. Several of the references are older but they are necessary as background support for the paper. Shoulder resurfacing has been evolving for many years.
Sincerely,
James W. Pritchett, MD, Author
Reviewer 2 Report
In the group 2-Total resurfacing arthroplasty with a ceramic humeral resurfacing component articulat- 93 ing with a polyethylene glenoid-it is not specified the etiology for shoulder resurfacing arthroplasty,just the number.There are with osteonecrosis,RA,primary Shouder OA etc?
For the others groups 1 and 3 the results are ok. This resurfacing hemiarthroplasty has a low revision rate with few post-operative.The number of complications was high se they need to improve the procedure.
The results are clearly presented and the advantage of use of ceramic component for shoulder resurfacing is a new concept.
Author Response
This reviewer had no comments or suggestions.
Round 2
Reviewer 1 Report
Thank you for your amendments. I strongly believe the paper is worth being published. Congratulations.